# Social Stigma, Mental Health, Stress, and Health-Related Quality of Life in People with Long COVID

**DOI:** 10.3390/ijerph20053927

**Published:** 2023-02-22

**Authors:** Urte Scholz, Walter Bierbauer, Janina Lüscher

**Affiliations:** 1Applied Social and Health Psychology, Department of Psychology, University of Zurich, Binzmuehlestrasse 14/Box 14, 8050 Zurich, Switzerland; 2University Research Priority Program “Dynamics of Healthy Aging”, University of Zurich, Stampfenbachstrasse 73l, 8006 Zurich, Switzerland; 3Swiss Paraplegic Research, Guido A. Zaech-Strasse 4, 6207 Nottwil, Switzerland

**Keywords:** long COVID, social stigma, depressive symptoms, anxiety, health-related quality of life, stress

## Abstract

A considerable amount of people who have been infected with SARS-CoV-2 experience ongoing symptoms, a condition termed long COVID. This study examined nuanced experiences of social stigma in people with long COVID and their associations with perceived stress, depressive symptoms, anxiety, and mental and physical health-related quality of life (hrqol). A total of *N* = 253 participants with long COVID symptoms (mean age = 45.49, SD = 12.03; *n* = 224, 88.5% women) completed a cross-sectional online survey on overall social stigma and the subfacets enacted and perceived external stigma, disclosure concerns, and internalized stigma. Data were analysed using multiple regression and controlling for overall burden of consequences of long COVID, overall burden of symptoms of long COVID, and outcome-specific confounders. In line with our preregistered hypotheses, total social stigma was related to more perceived stress, more depressive symptoms, higher anxiety, and lower mental hrqol, but—in contrast to our hypothesis—it was unrelated to physical hrqol after controlling for confounders. The three subscales of social stigma resulted in differential associations with the outcomes. Social stigma experiences go hand in hand with worse mental health in people with long COVID. Future studies should examine potential protective factors to buffer the effects of social stigma on people’s well-being.

## 1. Introduction

The adverse effects of the COVID-19 pandemic on societies and individuals are manifold. One set of consequences receiving more and more attention are the persistent residual effects of acute infection with SARS-CoV-2, termed “long COVID” or “post-acute COVID-19” [1]. As most studies on long COVID focus on medical questions around this new disease, more research on the psychosocial consequences of long COVID is urgently needed. This study addresses this research gap by focusing on social stigma experiences and their consequences for mental health, health-related quality of life, and perceived stress in people with long COVID.

### 1.1. Long COVID: Definition, Symptoms, and Prevalence

Long COVID or post-acute COVID is a new medical condition that is one of the consequences of an acute COVID-19 infection [1]. Different definitions of long COVID exist; for example, post-acute COVID-19 is diagnosed when symptoms persist four weeks beyond the onset of acute infection without other diagnoses explaining these symptoms [2]. Alternatively, the World Health Organization defines long COVID as symptoms still present three months or more after the onset of an acute infection and lasting for at least two months, again without an alternative diagnosis explaining the symptoms [3]. Numerous symptoms seem to be related to long COVID, with the most frequent being fatigue, headache, attention disorder, hair loss, and dyspnoea [4]. Reported prevalences vary greatly across studies. The World Health Organization estimates that between 10 and 20% of people who have an acute infection with SARS-CoV-2 will develop long COVID [1].

The vast majority of studies on long COVID focus on the medical aspects of the disease, but more and more studies are also taking the psychosocial consequences of living with long COVID into account. One of these psychosocial consequences is social stigma [5,6,7,8].

### 1.2. Long COVID and Social Stigma

Stigma was originally defined as a personal attribute people carry that leads to discriminative experiences [9]. Given that this definition seems to place responsibility on the people experiencing stigma, an alternative definition conceptualizes stigma as a social construction [10,11]. There are different forms of stigma experiences, including stressful life events [10] and secondary stigma experiences in relatives of stigmatized people [12]. In line with the finding that daily stressful experiences are highly relevant to people’s health [13], we focused on different everyday experiences involving social stigma based on the social stigma framework developed by Frost [10]: everyday discrimination, expectations of rejection, stigma management, and internalized stigma. Everyday discrimination, also often called enacted stigma [14,15], refers to the perceived negative social experiences people undergo due to their condition. Expectations of rejection, also called public attitudes [14] and perceived external stigma [15], refer to anticipated negative reactions from others. Stigma management includes concerns about disclosing the health issue [14,15]. Finally, internalized stigma refers to negative attitudes about oneself [10].

Social stigma is a common experience among people with various health conditions [11,12]. Researchers already reported on stigma experiences in the context of acute COVID-19 infections (e.g., [16,17,18]). Social stigma is also a common experience among people with unexplained medical symptoms [8]. The same applies for people with long COVID, as has been reported in both quantitative and qualitative research (for a qualitative review, see [7]). However, one research gap concerning social stigma experiences and long COVID is that nuanced analyses of associations between overall social stigma, as well as the different stigma experiences outlined above, and different indicators of mental health, stress, and health-related quality of life are so far lacking. Addressing this was one of the aims of this study.

The negative emotional consequences of social stigma experiences are manifold: experiences of social stigma contribute to perceived stress and negatively impact people’s health and quality of life [10,12]. In particular, experiencing social stigma has been reported to relate to enhanced stress [12,19,20]. Negative effects on people’s mental health, indicated by increased levels of depressive symptoms and anxiety, have also been reported in recent studies on acute COVID-19 (e.g., [20]). Aside from mental health and stress, people’s quality of life can also be significantly affected (cf. [12]). In the context of the social stigmatization of a health-related condition, health-related quality of life is the pertinent outcome. A common approach is to distinguish between mental and physical health-related quality of life [21,22]. 

It is important to note that outcomes of social stigma experiences are not universal but context- and population-specific [10,11]. For example, in the context of infectious diseases, such as acute COVID-19 or HIV, people might feel ostracized, which might lead to higher levels of social anxiety [20] but might not directly affect their physical health-related quality of life. There is a research gap with regard to these different outcomes of social stigma experiences in relation to long COVID: previous studies on social stigma experiences in persons with long COVID have primarily focused on descriptive questions; that is, the prevalence of social stigma experiences [23,24] or whether social stigma was among the emerging themes in reports of persons with long COVID [7,25,26]. Thus, given that there is relatively little knowledge about the psychosocial consequences of long COVID, we decided to examine various indicators of stress, mental health, and health-related quality of life in order to provide a comprehensive picture of associations between social stigma in relation to long COVID and its psychosocial consequences.

### 1.3. Aim of the Present Study

The aim of the present study was twofold: First, we examined the associations between social stigma experiences and various indicators of stress, mental health, and health-related quality of life in people with long COVID. For this, the following preregistered hypotheses (https://osf.io/ych3s) were tested: 

**H1:** 
*Higher levels of perceived long COVID-related stigma are related to more perceived stress.*


**H2:** 
*Higher levels of perceived long COVID-related stigma are related to more depressive symptoms and higher levels of anxiety.*


**H3:** 
*Higher levels of perceived long COVID-related stigma are related to lower levels of health-related quality of life.*


A second aim was to explore the associations between different subscales of stigma (enacted stigma, disclosure concerns, internalized stigma, external stigma) and the abovementioned outcomes. 

## 2. Materials and Methods

### 2.1. Participants and Study Design

This study was part of a larger project on the psychosocial consequences of long COVID. A detailed description of the larger project can be found at https://osf.io/sqjdu. Participants for this cross-sectional, correlational, online-survey study were recruited via social media (long COVID self-help groups on Facebook; Twitter), posts on online platforms, and mailings to a network of people with long COVID. Recruitment took place between June 2021 and October 2021 in Switzerland, Germany, and Austria. Inclusion criteria were being 18 years of age or older, understanding German, having had an acute COVID-19 infection 12 weeks before or longer, and experiencing persistent symptoms at the time of recruitment (long COVID). The Ethics Committee of the Faculty of Arts and Social Sciences of the University of Zurich reviewed and approved the study (reference number: 21.4.3). All participants signed an informed consent form.

### 2.2. Measures

Means and standard deviations of all measures are reported in Table 1. All item examples are translated from German. 

*Perceived social stigma* was assessed using a COVID-specific stigma questionnaire [15] that was adapted to long COVID. The adapted questionnaire consisted of 12 items with a six-point response format ranging from 0 (not true at all) to 5 (completely true). A total mean score for perceived stigma was calculated using all 12 items (Cronbach’s alpha = 0.89). For the subscales, exploratory factor analysis with varimax rotation revealed three factors (eigenvalue > 1) instead of four, as in the original version of the scale. The three items of *enacted stigma* (e.g., “It hurts me how people reacted when they learned that I have long COVID”) and the four items of *perceived external stigma* (e.g., “People with long COVID are treated as outcasts”) loaded on the same factor. Given the similarity between the two subfacets of (perceived) enacted stigma and perceived external stigma, these items were collapsed into one scale (Cronbach’s alpha = 0.90). Furthermore, the two items of the subscale *disclosure concerns* (e.g., “I pay close attention to whom I tell that I have long COVID”; Cronbach’s alpha = 0.65) and the three items of *internalized stigma* (e.g., “I have the feeling that I am not as good as others since I got long COVID”; Cronbach’s alpha = 0.70) were each averaged to build two different subscales. 

*Anxiety and depressive symptoms* were measured as indicators of mental health with the Hospital Anxiety and Depression Scale (HADS) [27]. The HADS consists of 14 items, with a response scale ranging from 0 to 3. A sum score for the seven items measuring anxiety was calculated (Cronbach’s alpha = 0.82). Moreover, the seven items measuring depressive symptoms were also combined into a sum score (α = 0.82). Higher scores indicated higher anxiety and more depressive symptoms.

*Perceived stress* was measured with the Perceived Stress Scale (PSS-4) [28], which included four items with a response format ranging from 0 (never) to 5 (very often) that were collapsed into a mean score. Cronbach’s alpha was 0.80. 

*Health-related quality of life* was measured with the SF-12 [21,22]. Two different composite scores for physical and mental health-related quality of life were built in accordance with the official coding scheme of the SF-12. As is common practice for the SF-12, composite scores were Z-transformed (*M* = 50 ± 10).

Furthermore, we assessed the perceived *overall burden of long COVID symptoms* with the question, “Overall, how burdensome have you perceived the disease with all its symptoms?” and the perceived *overall burden of consequences of long COVID* with the question, “Overall, how burdened have you been by the consequences of the long COVID disease?”, with a response format for both questions ranging from 0 (not at all burdened) to 10 (extremely burdened).

Moreover, standard *sociodemographic information* was assessed for all participants.

### 2.3. Data Analysis

Statistical analyses were conducted with SPSS 28 (IBM, Armonk, NY, USA). We ran bivariate correlations between all the main constructs of the study (see Table 2) and between the main constructs and the potential control variables age and sex (see Appendix A in the Appendix A). 

Our preregistered analysis plan (https://osf.io/ych3s) was to determine bivariate correlations and implement regression analyses, as well as sensitivity analyses to control for potential confounders, such as age, sex, and illness severity. We present the bivariate correlations in Appendix A. The regression analyses functioned, at the same time, as the sensitivity analyses and included the potential confounders in order to add new insights in addition to the bivariate associations already reported. In particular, we decided to control for overall burden of consequences of long COVID and overall burden of long COVID symptoms as indicators of illness severity in all analyses in order to identify the unique associations between social stigma and stress, depressive symptoms, anxiety, and health-related quality of life. Moreover, we controlled for depressive symptoms in the analyses with anxiety as the outcome and we controlled for anxiety in the analyses with depressive symptoms as the outcome in order to exclude the alternative explanation for the effects that they were driven by the overlap between depressive symptoms and anxiety. For the same reason, we controlled for physical health-related quality of life in the analyses with mental health-related quality of life as the outcome and vice versa. Finally, we added age and sex to the analyses of all outcomes that showed significant bivariate associations with these potential confounders. 

We conducted an a priori power analysis. Given that this is one of the first studies examining associations between social stigma and different outcomes in individuals with long COVID, we conservatively assumed small effect sizes. Power analysis was performed with G*Power for sample size estimation [29]. With a power of 0.90, a maximum of seven predictors (the different social stigma subscales plus control variables), an assumed effect size for the regression coefficients of 0.10, an alpha level of 0.05, and a two-tailed test, the required sample size would be *N* = 108. Thus, the study was sufficiently powered with a sample size of *N* = 253 participants.

## 3. Results

### 3.1. Sample Description

A total of *N* = 253 participants completed the questionnaire. Most participants were Swiss citizens (*n* = 142; 56.1%), *n* = 95 (37.5%) were German, *n* = 8 (3.2%) were Italian, *n* = 7 (2.8%) were Austrian, and 1 participant (0.4%) was a French citizen. Participants’ mean age was 45.49 (*SD* = 12.03; range = 20–83 years), with *n* = 224 (88.5%) being women. The majority (*n* = 166; 65.6%) were married or in a committed relationship, *n* = 52 (20.6%) of participants were single, *n* = 33 (13.1%) were divorced or widowed, and *n* = 2 (0.8%) did not respond. The majority of participants reported having a university degree (*n* = 129; 51%), while *n* = 111 (43.9%) had completed vocational training, *n* = 4 (1.6%) of participants were still undertaking their studies/vocational training, and *n* = 9 (3.6%) did not respond to this question. In terms of income, *n* = 18 (7.1%) reported having a total household income of less than CHF 2000 (approx. USD 2200), *n* = 27 (10.7%) reported having an income between CHF 2001 and 4000 (approx., USD 2201–4400), *n* = 50 (19.8%) reported a total household income between CHF 4001 and 6000 (approx. USD 4401–6500), *n* = 48 (19.0%) reported a total household income between CHF 6001 and 8000 (approx. USD 6501–8700), *n* = 31 (12.3%) reported a total household income between CHF 8001 and 10,000 (approx. USD 8701–11,000), *n* = 33 (13.0%) reported an income greater than CHF 10,000 (>approx. USD 11,000), *n* = 44 (17.4%) did not know or preferred not to say, and the responses of *n* = 2 (0.8%) were missing.

Overall, *n* = 227 (89.7%) of the participants had either tested positive for COVID-19 or had had the antibodies detected via a COVID-19 antibody test. Moreover, *n* = 205 (81%) participants reported that their long COVID condition had been confirmed by medical staff. The most frequently reported symptoms of long COVID were fatigue (*n* = 221, 87.4%), difficulties concentrating (“brain fog”; *n* = 196; 77.5%), and attention deficits (*n* = 187; 73.9%). The least reported symptoms were sleep apnoea (*n* = 27; 10.7%), loss of weight (*n* = 22; 8.7%), and pulmonary fibrosis (*n* = 8; 3.2%). The most reported pre-existing illnesses were diseases of the musculoskeletal system (*n* = 60; 23.7%) and psychological disorders (*n* = 29; 11.5%). Most participants contracted COVID-19 either in March 2020 (*n* = 40; 15.8%) or between October 2020 and March 2021 (*n* = 181; 71.5% in total). Concerning the subjective rating of acute COVID-19 symptom severity, *n* = 53 (20.9%) reported having had strong symptoms, *n* = 145 (57.3%) reported having had moderate symptoms, *n* = 53 (20.9%) reported having had mild symptoms, and *n* = 1 (0.4%) reported having had no symptoms. A total of *n* = 34 (13.4%) were hospitalized during acute COVID-19 infection. 

### 3.2. Descriptive Results

On average, participants in this study reported relatively low levels of social stigma experiences (see Table 1). The mean scores for depressive symptoms and anxiety indicate that, on average, the sample was above (for depressive symptoms) or approached (for anxiety) the cut-off score of 8+ for possible clinical presentation of depression and anxiety [30]. On average, participants reported that they were moderately stressed. The mean levels of physical and mental health-related quality of life demonstrate that participants with long COVID reported relatively low health-related quality of life. Moreover, the mean burdens of long COVID symptoms and of consequences resulting from long COVID were high. 

As displayed in Table 2, the social stigma total score and the different subfacets of social stigma correlated positively with depressive symptoms, anxiety, and perceived stress, mostly with medium effect sizes. Small negative associations emerged with mental health-related quality of life and physical health-related quality of life, with the exception of the social stigma subscales disclosure concerns and internalized stigma, which were unrelated to physical health-related quality of life. Moreover, all social stigma scales were positively related to overall burden of consequences of long COVID and overall burden of long COVID symptoms with small effects. Concerning the associations of the social stigma subscales and the different outcomes and burden indicators with age and sex (see Appendix A in the Appendix A), only three significant correlations emerged: internalized stigma (*r* = −0.19, *p* = 0.001) and overall burden of long COVID symptoms (*r* = −0.13, *p* = 0.05) were negatively correlated with age, indicating that older participants were less likely to report internalized stigma and a lower burden of long COVID symptoms than younger participants. Sex was negatively related to physical health-related quality of life (*r* = −0.14, *p* = 0.05), indicating that women were more likely than men to report lower physical health-related quality of life.

### 3.3. Results of the Regression Analyses

As hypothesized (H1), total social stigma was positively related to perceived stress when controlling for the effects of the burden of consequences resulting from long COVID and the burden of long COVID symptoms (see Table 3). A 1 unit increase in total perceived social stigma resulted in 0.24 units more perceived stress, everything else being constant. A total of 15% of the variance in perceived stress was explained, with total social stigma accounting for 8% of the explained variance. All three social stigma subscales (enacted and perceived external stigma, see Appendix A in the Appendix A; disclosure concerns, see Appendix A; internalized stigma, see Appendix A) also resulted in positive associations with perceived stress.

The analyses examining hypothesis two (H2) on the positive associations between total social stigma and depressive symptoms and anxiety confirmed H2. For depressive symptoms, a 1 unit change in total social stigma resulted in a significant increase of 0.85 units in depressive symptoms while controlling for anxiety, the burden of consequences resulting from long COVID, and the burden of long COVID symptoms (see Table 3). A total of 34% of the variance in depressive symptoms was explained, with total social stigma explaining 4%. In the analyses with the three stigma subscales, enacted and perceived external stigma (see Appendix A in the Appendix A) and internalized stigma (see Appendix A), but not disclosure concerns (see Appendix A), were positively related to depressive symptoms while controlling for anxiety and the two forms of burden from long COVID. 

For anxiety, a 1 unit change in total social stigma resulted in a significant 0.80 unit change in anxiety while controlling for depressive symptoms, the burden of consequences resulting from long COVID, and the burden of long COVID symptoms (see Table 3). Overall, 30% of anxiety was explained, with 3% by total social stigma. With regard to the three subscales of social stigma, enacted and perceived external stigma (see Appendix A in the Appendix A) was unrelated, but disclosure concerns and internalized stigma (see Appendix A in the Appendix A) were positively related to anxiety when controlling for depressive symptoms and the two forms of burden from long COVID.

For physical and mental health-related quality of life, total stigma was only positively related to mental health-related quality of life and not to physical health-related quality of life in the regression analyses. Thus, H3 had to be partially rejected. Only the confounders mental health-related quality of life and sex, as well as the burden of consequences resulting from long COVID and the burden of long COVID symptoms, were significantly negatively related to physical health-related quality of life, resulting in a total explained variance of 23%. With regard to the three stigma subscales, only enacted and perceived external stigma (see Appendix A in the Appendix A) were significantly negatively related to physical health-related quality of life while controlling for confounders, while disclosure concerns and internalized stigma were unrelated to physical health-related quality of life (see Appendix A in the Appendix A).

For mental health-related quality of life, another picture emerged: total social stigma was negatively related to mental health-related quality of life after controlling for physical health-related quality of life and the two forms of burden from long COVID (see Table 3). A 1 unit increase in total social stigma resulted in a 2.57 unit decrease in mental health-related quality of life with all confounders held constant. A total of 19% of the variance in mental health-related quality of life was explained, with 6% by total social stigma. All social stigma subscales also resulted in negative associations with mental health-related quality of life when controlling for the confounders (see Appendix A in the Appendix A).

Finally, given that *n* = 48 participants reported that their long COVID had not been officially confirmed by a physician, we ran sensitivity analyses by rerunning the analyses with only the sample of *n* = 205 participants who reported that their long COVID had been confirmed by a physician. The pattern of results was unchanged (see Appendix A in the Appendix A).

## 4. Discussion

This cross-sectional study examined the associations between social stigma experiences among people with long COVID and perceived stress, depressive symptoms, anxiety, and physical and mental health-related quality of life. The results confirmed two out of three of our preregistered hypotheses: higher reports of social stigma were related to more perceived stress, more depressive symptoms, higher anxiety, and lower mental health-related quality of life. Contrary to our assumptions, social stigma was unrelated to physical health-related quality of life after controlling for confounders.

Overall, mean social stigma levels were rather low, indicating that participants were not heavily confronted with social stigma. However, for those experiencing social stigma, the hypothesized detrimental associations with stress and lower mental health and mental health-related quality of life were confirmed. This aligns with previous research on the undesired associations between social stigma and stress, mental health indicators, and health-related quality of life [12,19,20]. This highlights the importance of raising awareness and educating others about the existence of long COVID in order to avoid stigmatization. More studies are needed that further examine from whom and in what situations people with long COVID experience social stigma. For example, as indicated in a recent qualitative study, people with long COVID who experienced social stigma from healthcare professionals reported losing trust in these professions (e.g., [25]). Another study reported social stigma experiences in the form of not being taken seriously and achieving a diagnosis [26]. When these kinds of experiences lead vulnerable people to turn away from the healthcare system, a worsening of both mental and physical health, as well as health-related quality of life, might result. Evidence-informed targeted interventions educating social stigma perpetrators about people with long COVID might be an effective means to reduce stigmatization in this population (cf. [12]).

The exploratory analyses with the social stigma subfacets in this study provided nuanced insights into the associations between social stigma experiences and the different outcomes. This is important because outcomes of social stigma experiences are context- and population-specific [10,11]. While most of the bivariate associations between the different stigma subfacets and the outcomes were significant (except for those of physical health-related quality of life with disclosure concerns and internalized stigma), several net effects turned out to be non-significant after controlling for confounders in the multiple regression analyses. In particular, the association of social stigma with depressive symptoms seemed to be mainly based on enacted and perceived external stigma and internalized stigma but not on disclosure concerns after controlling for confounders. For anxiety, it was only disclosure concerns and internalized stigma that displayed significant unique associations in the regression analyses, not enacted and perceived external stigma. Finally, despite the non-significant effect of total social stigma on physical health-related quality of life, the subfacet enacted and perceived external stigma displayed a negative association after controlling for confounders. These results highlight the benefit of considering not only overall social stigma but also social stigma subfacets in order to gain a more comprehensive understanding of social stigma experiences and their differentiated associations with various stress, mental health, and quality of life outcomes. Future studies can build on this research by examining what coping mechanisms optimally match the different social stigma experiences of people with long COVID and can buffer their negative effects. For example, internalized stigma could be addressed by supporting people with long COVID in attributing their social stigma experiences to the ignorance of the stigmatizers towards this relatively new disease instead of to themselves (e.g., [10]).

Another point to consider in future studies is that stigma experiences due to long COVID might not emerge in isolation. Instead, different kinds of stigma experiences across different domains—e.g., health-related stigma, such as for long COVID, and stigmatization due to age, sex, race, or socioeconomic status—could potentially co-occur and, if so, are likely to interact, potentially leading to more severe consequences in particularly vulnerable people [12]. This should be taken into account in future studies.

### Strengths and Limitations

This study has several strengths. For example, the nuanced assessment of social stigma, together with the assessment of a set of different stress, mental health, and health-related quality of life outcomes, produced a comprehensive picture of the associations under study. This is particularly important because experiencing social stigma differs for different populations [10,12]. Thus, this study contributes to a better understanding of social stigma experiences among people with long COVID. Moreover, the sample size of around 250 participants from a highly burdened population—e.g., as indicated by the high mean levels of burden from symptoms and burden from the consequences of long COVID—can be considered another strength of this study.

Aside from these positive aspects, several limitations need to be considered when interpreting the results. The cross-sectional nature of this study imposed several limitations. First, causality could not be inferred. As a consequence, an alternative explanation for the associations found in this study could be that worse mental health and quality of life could increase the likelihood of perceptions of social stigma. Second, social stigma and mental health outcomes are likely to interact dynamically in the form of a downward spiral [10]. For example, the negative social stigma experiences of a person with long COVID at the workplace can lead to lower work satisfaction and more work-related stress, which might both contribute to lower work performance, in turn leading to further social stigma experiences. These kinds of dynamics require longitudinal data and cannot be captured in cross-sectional research. Controlling for the overall burden of consequences of long COVID and the overall burden of long COVID symptoms was one attempt to mitigate the limitations of the cross-sectional data, but longitudinal studies are required to examine changes in associations.

Another limitation was that the measure of social stigma was adapted from the context of acute COVID-19 infections [15] but was not comprehensively validated. Proper validation of measures takes time and also requires large sample sizes of the target population [31]. There seem to be attempts to develop validated measures of stigma experiences among people with long COVID [23]. Future studies should consider using these newly developed validated measures. However, the fact that the results mostly confirmed the hypothesized associations between the total social stigma score and the three subscales with the different outcomes speaks to the predictive validity of the social stigma measure used in this study.

This sample comprised a substantial number of participants who reported that their long COVID had not been officially confirmed by a physician. At the time of this study, there were long waiting lists for specialized consultations for long COVID in Switzerland, Germany, and Austria. Thus, we did not want to limit participation to only people with an official confirmation of their long COVID. However, there might also have been other reasons for not receiving this official confirmation by a physician. This is why we ran sensitivity analyses to examine the robustness of the results when excluding all participants without confirmation from a physician. The results were robust, thus indicating that having an official confirmation of long COVID did not make a difference for the associations between social stigma, including the subfacets, and the various outcomes considered in this study.

Finally, online surveys come with specific challenges, such as lack of control over who completes the survey and inattentiveness [32]. Overall, however, many studies attest that the quality of online surveys is comparable to on-site surveys [32]. Moreover, given that one of the major symptoms of long COVID is fatigue, this study might not have been possible if participants had been requested to travel to the university to complete the survey on-site. It would also not have been possible to reach people with long COVID from different German-speaking countries, as was achieved in this study.

## 5. Conclusions

Social stigma experiences went hand in hand with greater perceived stress, more depressive symptoms, higher anxiety, and lower mental health-related quality of life in people with long COVID while controlling for the burden of overall consequences and the burden of symptoms of long COVID and outcome-specific confounders. Moreover, considering different subfacets of social stigma, such as enacted and perceived external stigma, disclosure concerns, and internalized stigma, provided a nuanced picture of different aspects of social stigma and various stress, mental health, and health-related quality of life outcomes in this cross-sectional study with people with long COVID.

## Figures and Tables

**Table 1 ijerph-20-03927-t001:** Descriptive statistics for the main constructs of this study.

Psychological Measures	*M*	*SD*	Observed Range	*n*
Total social stigma ^1^	1.47	1.06	0–4.92	253
Enacted and perceived external stigma	1.62	1.27	0–5	253
Disclosure concerns	1.47	1.39	0–5	253
Internalized stigma	1.11	1.15	0–5	252
Depressive symptoms	8.58	3.99	1–21	253
Anxiety	7.92	4.09	0–19	253
Perceived stress	2.16	0.81	0–3.67	253
Physical quality of life	37.37	9.31	18.12–61	246
Mental quality of life	35.54	9.98	9.55–59.93	246
Overall burden of long COVID symptoms	8.55	1.72	0–10	235
Overall burden of consequences of long COVID	7.62	2.58	0–10	253

^1^ The social stigma total score was calculated as the mean score of all 12 stigma questionnaire items.

**Table 2 ijerph-20-03927-t002:** Correlations between total social stigma score, social stigma subscales, outcome measures, and overall burdens of consequences and symptoms.

Variable	1.	2.	3.	4.	5.	6.	7.	8.	9.	10.
1. Total social stigma	--									
2. Enacted and perceived external stigma	0.94 **	--								
3. Disclosure concerns	0.71 **	0.52 **	--							
4. Internalized stigma	0.69 **	0.45 **	0.45 **	--						
5. Depressive symptoms	0.44 **	0.37 **	0.28 **	0.45 **	--					
6. Anxiety	0.42 **	0.30 **	0.36 **	0.48 **	0.55 **	--				
7. Perceived Stress	0.38 **	0.33 **	0.23 **	0.34 **	0.47 **	0.43 **	--			
8. Physical quality of life	−0.14 *	−0.20 **	−0.04	0.03	−0.14 *	0.15 *	−0.07	--		
9. Mental quality of life	−0.29 **	−0.21 **	−0.22 **	−0.33 **	−0.58 **	−0.53 **	−0.42 **	−0.22 **	--	
10. Overall burden of consequences of long COVID	0.23 **	0.20 **	0.14 *	0.19 **	0.22 **	0.09	0.28 **	−0.24 **	−0.18 **	--
11. Overall burden of long COVID symptoms	0.28 **	00.28 **	0.13 *	0.20 **	0.22 **	0.17 **	0.16 *	−0.32 **	−0.17 **	0.34 **

* *p* < 0.05. ** *p* < 0.01.

**Table 3 ijerph-20-03927-t003:** Regression analyses with total social stigma as predictor and confounders to predict the various outcomes.

	Perceived Stress ^a^				Depressive Symptoms ^b^				Anxiety ^c^				Physical Quality of Life ^d^				Mental Quality of Life ^e^			
	*beta*	*b*	*b* 95% CI		*beta*	*b*	*b* 95% CI		*beta*	*b*	*b* 95% CI		*beta*	*b*	*b* 95% CI		*beta*	*b*	*b* 95% CI	
			LL	UL			LL	UL			LL	UL			LL	UL			LL	UL
Intercept		0.87 **	0.35	1.38		1.63	−0.56	3.83		2.96 *	0.68	5.24		71.80 **	63.69	79.92		65.21 **	55.72	74.71
Total social stigma	0.23	0.24 **	0.14	0.34	0.22	0.85 **	0.40	1.31	0.20	0.80 **	0.32	1.28	−0.11	−1.04	−2.21	0.14	−0.26	−2.57 **	−3.79	−1.36
Overall burden of disease	0.21	0.07 **	0.03	0.11	0.13	0.20 *	0.03	0.37	−0.07	−0.11	−0.29	0.065	−0.19	−0.70 **	−1.15	−0.26	−0.15	−0.58 *	−1.07	−0.10
Overall burden of symptoms	0.01	0.003	−0.06	−0.07	0.04	0.10	−0.17	0.35	0.04	0.10	−0.18	0.37	−0.28	−1.53 **	−2.21	−0.84	−0.16	−0.91 *	−1.68	−0.14
Anxiety					0.42	0.41 **	0.30	0.52												
Depressive symptoms									0.45	0.45 **	0.33	0.58								
Sex ^f^													−0.13	−3.98 *	−7.64	−0.32				
Mental hrqol ^g^													−0.33	−0.30 **	−0.42	−0.19				
Physical hrqol ^g^																	−0.34	−0.36 **	−0.50	−0.23

Note. * *p* < 0.05. ** *p* < 0.01. Empty rows indicate that the covariate was not included in the analysis of this specific outcome. ΔR^2^ is the added explained variance for the total social stigma predictor after controlling for covariates. ^a^ adjusted R^2^ = 0.15; ΔR^2^ = 0.08. ^b^ adjusted R^2^ = 0.34; ΔR^2^ = 0.04. ^c^ adjusted R^2^ = 0.30; ΔR^2^ = 0.03. ^d^ adjusted R^2^ = 0.23; ΔR^2^ = 0.01. ^e^ adjusted R^2^ = 0.19; ΔR^2^ = 0.06. ^f^ 0 = male, 1 = female. ^g^ hrqol = health-related quality of life.

## Data Availability

The data and code presented in this study are openly available from OSF at https://doi.org/10.17605/OSF.IO/M2USJ (accessed on 1 February 2023).

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
