# Peer review of "Social Stigma, Mental Health, Stress, and Health-Related Quality of Life in People with Long COVID"

_ijerph, 2023, doi:10.3390/ijerph20053927_

Round 1
Author Response
The study gives clear picture of the experiences of the those suffering from Long COVID symptoms regarding its impact on their health related quality of life, social stigma, mental wellbeing and stress. The work contributes to literature on psychological consequences of Lon COVID.
Response: We thank the reviewer for their positive feedback, the careful reading and the constructive comments on how to improve our paper.
The following comments are made to improve the quality of the study for publication.
Abstract
The i.e. is not suitable as Long COVID has been considered by WHO. Rather use bracket (Long COVID) or restructure the sentence.
Response: Thank you. We now rewrote the sentence to “A considerable amount of people who were infected with SARS-CoV-2 experience ongoing symptoms, a condition termed Long-COVID.”
Conclusion not indicated.
Response: Thanks. We now added a conclusion to the abstract: “Social stigma experiences go hand in hand with worse mental health in people with Long-COVID. Future studies should examine potential protective factors for buffering the effects of social stigma on people’s well-being.»
Introduction
1.2. the sentence that starts with social stigma is not clear and looks incomplete. Please restructure.
Response: Thank you. We now changed the sentence to “Social stigma is a common experience in people with various health conditions [11,12].”
The literature gap for on this study is not clearly presented by the study.
Response: Thank you, we now provided clear information on the research gaps that our study addressed in part 1.2.
“However, one research gap concerning social stigma experiences and Long-COVID is that nuanced analyses of associations between overall social stigma as well as the different stigma experiences outlined above with different indicators of mental well-being, stress, and health-related quality of life (hrqol) are so far lacking. Addressing this was one of the aims of this study.”
and
“It is important to note that outcomes of social stigma experiences are not universal but context- and population-specific [10, 11]. For example, in the context of infectious diseases, such as acute COVID-19 or HIV, people might feel ostracized which might lead to higher levels of social anxiety [20], but might not directly affect their physical health-related quality of life. With regard to these different outcomes of social stigma experiences, there is a research gap for Long-COVID: Previous studies on social stigma experience in persons with Long-COVID have primarily focused on descriptive questions, that is, the prevalence of social stigma experiences [23, 24] or if social stigma was among the emerging themes in reports of persons with Long-COVID [7, 25, 26]. Thus, given that there is relatively little knowledge on the psychosocial consequences of Long-COVID, we decided to examine various indicators of stress, mental health, and health-related quality of life in order to allow providing a comprehensive picture of associations between Long-COVID and psychosocial consequences.”
Literature on depressive symptoms, health related quality of life and anxiety concepts lacking whereas the concepts appear under preregistered hypotheses and title.
Response: Thank you. We now added more information on the different outcomes and on the rationale for examining these in our study. The paragraph now reads:
“The negative emotional consequences of social stigma experiences are manifold: Experiences of social stigma contribute to perceived stress and negatively impact people’s health and their quality of life [10, 12]. In particular, experiencing social stigma has been reported to relate to enhanced stress [12, 19, 20]. Negative effects on people’s mental health, indicated by increased levels of depressive symptoms and anxiety, have also been reported in recent studies on acute COVID-19 [e.g. 20] from mental health and stress, people’s quality of life can also be significantly affected [cf. 12]. In the context of social stigmatization of a health-related condition, health-related quality of life is the outcome of choice. A common approach is to distinguish between mental and physical health-related quality of life [21, 22].
It is important to note that outcomes of social stigma experiences are not universal but context- and population-specific [10, 11]. For example, in the context of infectious diseases, such as acute COVID-19 or HIV, people might feel ostracized which might lead to higher levels of social anxiety [20], but might not directly affect their physical health-related quality of life. With regard to these different outcomes of social stigma experiences, there is a research gap for Long-COVID: Previous studies on social stigma experience in persons with Long-COVID have primarily focused on descriptive questions, that is, the prevalence of social stigma experiences [23, 24] or if social stigma was among the emerging themes in reports of persons with Long-COVID [7, 25, 26]. Thus, given that there is relatively little knowledge on the psychosocial consequences of Long-COVID, we decided to examine various indicators of stress, mental health, and health-related quality of life in order to allow providing a comprehensive picture of associations between Long-COVID and psychosocial consequences..”
Methodology
The authors should consider indicating the number of participants (n) from each country as the first in sample description.
Response: As requested by the reviewer, we now reported the country of origin right after the total number of participants in the sample description.
Discussion: the authors should include more literature to the discussion.
Response: Thank you. We now added more references to the discussion.
Reviewer 2 Report
Dear authors,
congratulations on very interesting study.
This study aimed to investigate the relationship of perceived social stigma in people who have Long COVID symptoms with perceived stress, depressive symptoms, anxiety, and physical and mental health-realted quality of life.
Introduction is well written and sufficiently does its job of introducing the reader into the study specific objectives. I would maybe refrain from using the abbreviation hrql for the health-realted quality of life throughout the entire manuscript. Maybe leave it in tables, but in the text, use full version.
I have few concerns related to the Materials and Methods section in particular. Firstly, large parts of the sample description should be moved to the Results section including Tables 1 and 2. Leave only pure description of your sample and measures in this section. Everything else came as a result of your collected and later processed data.
You reported 253 participants, but later you mention that only 227 of them were tested positive, and 205 of them have been officially diagnosed with long COVID. Hence, you cannot claim that you have recruited 253 participants with long COVID. It is bit inaccurate. Your inclusion criteria was having acute COVID-19 infection 12 weeks or longer and still experiencing symptoms. It is important to distinguish those who have been truly diagnosed with COVID-19 and long COVID-19 from those who have self-diagnosed themselves.
Please include sociodemographic questionnaire into measures section. From your sample description section it is evident that you have have collected that type of info.
Minor:
Table 3- you jump from superscript letter c (anxiety) to e (physical quality of life). Please modify this.
In my opinion, limitation section should include downsides of online survey study and unclear (or unsure) COVID-19/long COVID-19 diagnosis.
Overall, this study is very well executed and results are novel.
If you have any questions, please do not hesitate to ask.
Author Response
Dear authors, congratulations on very interesting study. This study aimed to investigate the relationship of perceived social stigma in people who have Long COVID symptoms with perceived stress, depressive symptoms, anxiety, and physical and mental health-realted quality of life.
Response: We thank the reviewer for their positive feedback, the careful reading and the constructive comments on how to improve our paper.
Introduction is well written and sufficiently does its job of introducing the reader into the study specific objectives. I would maybe refrain from using the abbreviation hrql for the health-realted quality of life throughout the entire manuscript. Maybe leave it in tables, but in the text, use full version.
Response: Thank you. We now replaced the abbreviation by the full term as suggested by the reviewer except in the abstract and tables.
I have few concerns related to the Materials and Methods section in particular. Firstly, large parts of the sample description should be moved to the Results section including Tables 1 and 2. Leave only pure description of your sample and measures in this section. Everything else came as a result of your collected and later processed data.
Response: We now moved the sample description and Tables 1 and 2 to the results section.
You reported 253 participants, but later you mention that only 227 of them were tested positive, and 205 of them have been officially diagnosed with long COVID. Hence, you cannot claim that you have recruited 253 participants with long COVID. It is bit inaccurate. Your inclusion criteria was having acute COVID-19 infection 12 weeks or longer and still experiencing symptoms. It is important to distinguish those who have been truly diagnosed with COVID-19 and long COVID-19 from those who have self-diagnosed themselves.
Response: We thank the reviewer for this critical and constructive comment. We did not limit participation to people already being officially diagnosed with Long-COVID because in Switzerland there are long waiting lists for people with this condition. We agree with the reviewer, however, that being officially diagnosed or not could make a difference with regard to the results. Thus, we ran sensitivity analyses in selecting only those who reported that Long-COVID was confirmed by a physician. The pattern of results remained unchanged in that all significant or non-significant effects of the total stigma score as well as the subscales were replicated. The effects were in parts a bit stronger, but not substantially. We now added this to the results section and added all results of the analyses with this subsample to the supplemental material. The sentence in the results section now reads: “Finally, given that n = 48 participants reported that their Long-COVID had not been officially confirmed by a physician, we ran sensitivity analyses by rerunning the analyses only with the sample of n = 205 participants who reported that their Long-COVID had been confirmed by a physician. The pattern of results was unchanged (see Tables S5-S8 in the supplement).”
Please include sociodemographic questionnaire into measures section. From your sample description section it is evident that you have have collected that type of info.
Response: We now added a sentence that we also assessed sociodemographic information from the participants to the measures section. We also added to the methods section that this study was part of a larger project and provided an OSF-Link for more information on the larger project.
Minor:
Table 3- you jump from superscript letter c (anxiety) to e (physical quality of life). Please modify this.
Response: Thank you for spotting this. We corrected the superscript.
In my opinion, limitation section should include downsides of online survey study and unclear (or unsure) COVID-19/long COVID-19 diagnosis.
Response: We now added these aspects to the limitations section:
“This sample comprised a substantial number of participants who reported that their Long-COVID had not been officially confirmed by a physician. At the time of this study there were long waiting lists for specialized consultations for Long-COVID in Switzerland, Germany, and Austria. Thus, we did not want to limit participation to people with an official confirmation of their Long-COVID only. However, there might also be other reasons for not having received this official confirmation by a physician. This is why we ran sensitivity analyses in order to examine the robustness of the results when excluding all participants without a confirmation by a physician. Results were robust, thus, indicating that having an official confirmation of Long-COVID or not did not make a difference for the associations between social stigma, including the subfacets, and the various outcomes considered in this study.
Finally, online surveys come with specific challenges, such as lack of control of who completes the survey or inattentiveness [32]. Overall, however, many studies attest that the quality of online surveys is comparable to one-site ones [32]. Moreover, given that one of the major symptoms of Long-COVID is fatigue, this study might not have been possible if participants had been requested to travel to the University to complete the survey on-site. It would also not have been possible to reach people with Long-COVID from different German-speaking countries, as was the case in this study.”
Overall, this study is very well executed and results are novel. If you have any questions, please do not hesitate to ask.
Response: Thank you!